# Associations between Perception of Help-Seeking Behaviors and Quality of Life among Older People in Rural Communities: A Cross-Sectional Study

**DOI:** 10.3390/ijerph192013331

**Published:** 2022-10-15

**Authors:** Ryuichi Ohta, Chiaki Sano

**Affiliations:** 1Community Care, Unnan City Hospital, 96-1 Iida, Daito-Cho, Unnan 699-1221, Japan; 2Department of Community Medicine Management, Faculty of Medicine, Shimane University, 89-1 Enya Cho, Izumo 693-8501, Japan

**Keywords:** help-seeking behaviors, quality of life, rural community, EQ-5D-5L, Japan, older people

## Abstract

Older people’s help-seeking behaviors (HSBs) may be limited because of various factors and are essential in improving healthcare in aging societies. This cross-sectional study investigated the association between perception of HSBs, concrete HSBs, quality of life (QOL), and other variables among people over 65 in rural Japan using standardized questionnaires. Participants were divided into high or low health status index score groups based on a median split. Logistic regression was used to assess the association between perception of HSBs and high QOL while controlling for age, sex, living conditions, annual health checks, having chronic diseases, regular clinic visits, smoking, habitual alcohol consumption, education, living conditions, social support, social capital, socioeconomic status (SES), and health literacy. Participants in the high QOL group were younger (*p* < 0.001), and had fewer chronic diseases and regular clinic visits than those in the low QOL group (*p* < 0.001). The multivariate logistic regression model revealed that age, chronic diseases, tobacco usage, family consultation, and consulting primary care physicians negatively predicted QOL. High SES, social capital and support, and HSB intention positively predicted QOL. Self-efficacy and intention regarding HSBs should be investigated to improve health among older rural people.

## 1. Introduction

Help-seeking behaviors (HSBs) are behaviors people engage in when experiencing certain symptoms and health difficulties [1]. There are different HSBs related to (acute and chronic) symptoms and conditions and various demographic characteristics [2,3,4]. HSBs are categorized as lay and professional care, differentiated by whether one uses professional healthcare care [5]. Social factors such as age, gender, living location, and relationships with others in the community affect the tendency to use lay or professional care [6,7,8]. HSBs can affect people’s health conditions, such as self-rated health and quality of life (QOL) [6,9,10]. Hence, it is important to consider HSBs to improve healthcare.

Older people’s HSBs are a pressing issue when considering the improvement of health care in aging societies. Older people’s HSBs may be limited compared with younger generations because of the low mobility and availability of healthcare resources [11,12,13]. Older people’s reduced access to healthcare resources could affect their health status [14]. Moreover, the increase in older people who cannot use cars and transportation systems due to frailty has hindered the current healthcare systems [15]. Providing support for older people’s HSB is vital for comprehensive care in communities as this leads to improved QOL.

Older people face more challenges in HSBs in rural areas because of the lack of resources compared with urban areas and need effective HSBs. The outflow of the younger generation and the increase in the rate of the older population in rural areas have reduced the accessibility of health care resources for the latter without younger people’s help [16,17]. Moreover, older people tend to have multiple medical problems, known as multimorbidity and require professional care [18]. Medical care usage has increased, causing polypharmacy (i.e.,) the continuous use of multiple medicines for chronic diseases [19]. Multimorbidity and polypharmacy could trigger various symptoms in older people and require lay and professional care [19,20,21]. Older adults must develop effective HSBs for their usual symptoms to have good QOL.

Changes in existing HSBs can help improve QOL because various HSBs are related to it. Previous research reveals that self-management of usual symptoms may be related to higher QOL among older people in the rural Japanese context [22,23]. Similarly, self-medication (i.e.,) using various over-the-counter drugs and home remedies is also associated with high QOL [9]. However, the need for family support and frequent medical care may be related to low QOL [24,25]. In rural contexts, HSBs, including self-management and self-medication, could improve older people’s QOL.

According to social cognitive theory, effective health behaviors require older people to have self-efficacy and the intention to exhibit HSBs [26,27,28]. Social cognitive theory is essential for understanding HSBs through the interactions between people, contexts, and behaviors. Self-efficacy is an individual’s belief in their capacity to perform HSB successfully, leading to the intention of the behaviors. Self-efficacy and intention are important prerequisites for HSBs [26,27,28]. Per the theory, knowledge and understanding of healthy behaviors alone may not drive effective behaviors and high self-efficacy and intention are possible drivers of such behaviors. This theory explains the process of change in HSBs as one of the health-related behaviors [26,27,28]. However, with respect to the social cognitive model, existing evidence does not clarify the relationship between rural people’s perceptions of HSBs and QOL, in relation to self-efficacy and intentions. Therefore, this study aimed to answer the research question: “Is the perception of HSBs, among rural older people, associated with high QOL?” Clarifying the association between the perception of HSBs and QOL among the older rural population can effectively change the focus of interventions related to HSBs from people’s behaviors to their perception of HSBs. The content of the interventions can then focus on older people’s self-efficacy and intention to perform HSBs, rather than only behaviors. Thus, this study aimed to investigate the association between the perception of HSBs and QOL among older people in rural Japan.

## 2. Materials and Methods

### 2.1. Setting

This research was performed in Kakeya and Yoshida in Unnan City, located in southeast Shimane prefecture, Japan. It is primarily covered by forests and is one of the most rural cities in Japan. A 2017 survey revealed that the total population of Unnan City was 38,882 (18,720 males and 20,162 females), with an aging rate of 37.82%, estimated to reach 50% by 2025 [29]. Each family lives separately. Among the six towns of Unnan city, Kakeya and Yoshida lie in the extreme southwestern part. They have two clinics each and are far from a general hospital, which is more than a 30 min drive. They are thus different from other towns which have multiple clinics and hospitals. In total, Kakeya and Yoshida towns have six regions, among which participants from four regions (Kakeya, Tai, Matsukasa, and Tane) agreed to participate in the research.

### 2.2. Participants

Overall, 572, 247, 129, and 211 people over 65 years resided in Kakeya, Tai, Matsukasa, and Tane, respectively. These regions are neighboring districts. Participants were briefed about this study through a letter comprising its details and a research questionnaire. Patients who could not adequately read or write and those with dementia were excluded from this study.

### 2.3. Measurements

A questionnaire was administered to all participants. Community workers in each region distributed the questionnaire to each participant and collected it once completed.

#### 2.3.1. Assessment of Perception of HSBs

To measure participants’ perception of HSBs, we used a model of human behavior change based on the social cognitive theory [26,27]. According to this model, people change their behavior in four steps: knowledge, understanding, self-efficacy, and intention [26,27]. The more knowledge people have about a particular behavior and the more deeply they understand it, the more interest and self-efficacy they have, leading to intention and the actual behavior. We asked about knowledge, understanding, self-efficacy, and intention regarding HSBs in relation to usual symptoms, using the following questions: (a) Do you know how to deal with your usual symptoms? (Knowledge) (b) Do you understand how you should deal with your usual symptoms? (Understanding) (c) Do you feel you can deal with your usual symptoms? (Self-efficacy) (d) Do you intend to deal with your usual symptoms? (Intention) [30]. Participants completed these questions on a four-point Likert scale (agree, agree a little, disagree a little, disagree). The definition of usual symptoms involved symptoms not causing concern, usually indicative of a trivial/self-limiting illness. Based on these findings, the participants were presented with examples of mild symptoms, including mild fatigue, flu-like symptoms, joint pain, back pain, and mild headache. Each questionnaire was categorized binomially (high (agree or agree a little) = 1, low (disagree a little or disagree) = 0).

#### 2.3.2. Assessment of HSBs for Usual Symptoms

A validated questionnaire assessed patients’ preference for lay care for mild symptoms [5]. In this questionnaire, participants chose their behavioral tendencies when they had mild symptoms. Care choices included doing nothing, self-management (observing, sleeping, resting, and taking a bath), seeking information, consulting family, friends, or community members, using complementary or home medicines, buying over-the-counter drugs, consulting pharmacists or primary care physicians, visiting medical institutions (other than primary care physicians), and visiting the emergency rooms of general hospitals (including calling an ambulance). Based on previous literature, we used self-management, consulting family, and self-medication, including using complementary or home medicines, buying over-the-counter drugs, and consulting primary care physicians as variables relating to QOL [6,9,22,23,24].

#### 2.3.3. QOL Measurement

As a primary outcome, QOL was measured as health status index scores using the EuroQOL (EQ-5D-5L) questionnaire. It is a self-administered questionnaire and descriptive system comprising five dimensions of health status (mobility, self-care, usual activities, pain/discomfort, and anxiety/depression) [31]. The health status of each dimension is measured using five levels of severity: no problems, slight problems, moderate problems, severe problems, and extreme problems. The responses to the five dimensions are combined into a five-digit number showing the participant’s health status profile (from “11111,” meaning no problems at all, to “55555,” showing extreme problems in all five dimensions). In total, 3125 possible health conditions are defined in this manner. Health status profiles are converted into a single health status index score by applying a formula that attaches values to each response [32]. This questionnaire has been validated in various languages, including Japanese [32]. This study used the Japanese version.

#### 2.3.4. Other Confounders

The questionnaire items included, age, sex, body mass index (BMI), having chronic diseases, regular clinic visits, smoking, habitual alcohol consumption, educational level, living conditions, social support [33], social capital (using a 10-point Likert scale ranging from “can rely on neighbors in communities” to “cannot completely rely on neighbors in communities”) [34], and socioeconomic status (SES). The independent variables were categorized binomially: sex (male = 1, female = 0), smoking (yes = 1, no = 0), habitual alcohol drinking (yes = 1, no = 0), educational level (more than graduation from high school = 1, no = 0), living condition (with family = 1, residing alone = 0), social support (present or relatively present = 1, absent or relatively absent = 0), social capital (high (10 to 6) = 1, low (5 to 1) = 0), SES (high (rich or relatively rich) = 1, low (poor or relatively poor) = 0).

### 2.4. Statistical Analyses

Parametric and categorical data were analyzed using Student’s *t*-test and chi-squared test, respectively. A significance level of *p* < 0.05 was used for all comparisons. Participants were divided into two groups depending on whether their health status index scores were above or below the median. The median was 0.705, with an interquartile range of 0.592 to 0.814. To investigate the statistical difference in the rate of participants with a high intention for HSB in the two groups, a minimum of 219 participants were required in each group based on α (alpha) = 0.05, β (beta) = 0.20, and a power of 80% for a difference in the rate of 0.1.

A logistic regression model was used to assess the association between the perception of HSBs and high QOL, controlling for the potential confounding factors of age, sex, living conditions, annual health checks, having chronic diseases, regular clinic visits, smoking, habitual alcohol consumption, educational level, living conditions, social support, social capital, SES, and health literacy. All statistical analyses were performed using Easy R (Saitama Medical Center, Jichi Medical University, Saitama, Japan), a graphical user interface for R (The R Foundation, Vienna, Austria) [35].

### 2.5. Ethical Considerations

Participants were informed that the data collected would be used only for research purposes. Furthermore, they were informed about the research objectives, how their data would be disclosed, and the protection of their personal information. Subsequently, they provided written consent. This study was conducted following the principles of the Declaration of Helsinki. This study was approved by the Unnan City Hospital Clinical Ethics Committee (approval number: 20200013).

## 3. Results

### 3.1. Demographic Data

Figure 1 shows the flowchart of the participant selection process. The total population of Kakeya, Matsukasa, Tane, and Tai was 2694. Among them, 1159 were over 65 years. We included 1066 participants in total. Of these, 232 were excluded due to the missing responses and missing data regarding the EQ-5D-5L, SES, education, and social capital data (Figure 1).

The total effective response rate of the questionnaires was 78.2% (834/1066), and those of the participants from Kakeya, Tai, Matsukasa, and Tane were 80% (443/554), 67.9% (155/230), 97.2% (105/109), and 75.7% (131/173), respectively. Participants in the high QOL group were younger than those in the low QOL group (*p* < 0.001). Fewer participants in the former group had chronic diseases and regular clinic visits than those in the latter group (*p* < 0.001). Participants in the high QOL group had higher education levels than those in the other groups (*p* < 0.001). Additionally, more participants in the former group had a higher SES, social capital, social support, and annual checks and alcohol consumption rates than those in the latter (*p* < 0.001). No significant differences were found in sex, BMI, tobacco use, living with family, or social capital between the groups. Regarding the perception of HSBs, participants in the high QOL group had a higher rate of knowledge, understanding, self-efficacy, and intention of HSBs than the other group. Regarding concrete behaviors, participants in the high QOL group had a higher rate of self-management and self-medication than the other groups. Contrarily, participants in the high QOL group had a lower rate of family consultation and usage of primary care doctors than those in the other groups (Table 1).

### 3.2. Association between Perception of Help-Seeking Behaviors and High Quality of Life

A multivariate regression model was used to investigate the association between the perception of HSBs and high quality of life. This study referred to the previous research and used a logistic regression model with forced entry by including all previous independent variables. The C-statistic for logistic regression model was 0.807 (95% confidence interval [CI], 0.778–0.836). In the results of the multivariate logistic regression model, age (odds ratio [OR] = 0.90, 95% CI: 0.88–0.92), having chronic diseases (OR = 0.51, 95% CI: 0.30–0.86), tobacco usage (OR = 0.53, 95% CI: 0.28–1.00), family consultation (OR = 0.53, 95% CI: 0.37–0.76), and consulting primary care physicians (OR = 0.65, 95% CI: 0.44–0.97) were negatively associated with higher QOL. High SES (OR = 2.37, 95% CI: 1.68–3.34), social capital (OR = 1.96, 95% CI: 1.23–3.13), social support (OR = 1.81, 95% CI: 1.10–2.96), and intention for HSBs (OR = 2.95, 95% CI: 1.83–4.75) were positively associated with QOL. Other factors were not associated with QOL (Table 2).

## 4. Discussion

This study showed the association between HSB intentions regarding usual symptoms and high QOL among older rural people. The association persisted after adjusting for factors previously shown to be related to the QOL, including social support and capital, SES, and concrete HSBs. For better health conditions, changes in HSBs alone may not be sufficient. Perceptions of HSBs should be investigated to improve health conditions among older rural people.

This study is the first to show a positive association between HSB intention regarding usual symptoms and high QOL among older adults in rural areas. This finding can contribute to the development of interventions focusing on self-efficacy and intentions of HSBs among older adults. Several studies have shown that HSBs may be related to QOL and other health outcomes, such as self-rated health and longevity [6,9,22,23,24]. Previous studies suggest that changes in HSBs among older people could effectively improve their health conditions [23]. However, this study revealed that after controlling for perceptions regarding HSBs, concrete behaviors were not associated with QOL. This result indicates that concrete HSBs based on people’s perceptions of how they can behave when experiencing certain symptoms are essential. This can be explained by the social cognitive theory. According to the theory, imposed behaviors and help may not change people’s health conditions. Rather self-efficacy and intention of health behaviors should drive people’s behaviors and are essential [26,27,28]. Changes in HSBs alone may not change older people’s health conditions. Therefore, enforcing changes in their HSBs without improving their self-efficacy and HSB intentions may not change their health conditions. According to the social cognitive model concerning the modification of human behaviors, older adults should be motivated to act based on their perceptions of knowledge, understanding, self-efficacy, and intentions [26].

There is a possibility of a positive relationship between the intention to perform HSB and QOL. As this study shows, the intentions to perform HSBs such as approaches to usual symptoms could be associated with high QOL after adjusting for other social and health conditions. Therefore, the relationships among them can be considered in various ways. People with high QOL could have high motivation and intention regarding HSBs because they can move and act without subjective difficulty in their lives and have high social capital and support, and SES [36,37,38]. However, this study showed that the association between high intention to perform HSBs and QOL persisted after adjustment for these social and health conditions. Based on behavioral theories, the perception of health conditions, self-efficacy, and confidence in HSBs could lead to high QOL [39,40]. Although this study only showed an association between intention to perform HSBs and QOL, it suggests a positive relationship between HSB intention and high QOL based on behavioral theories.

For effective intervention in HSBs among older people, systems approaches should be implemented regarding cultural and social conditions. Systems approaches can be effective in public health interventions using various dimensions, such as personal, interpersonal, community, and organizational [41,42,43]. Present interventions regarding HSBs may mainly be provided by organizations such as governments. Changing medical insurance systems may restrict HSBs among older people [44,45]. In rural contexts, as this article shows, although educational levels may not be high, social support and capital are, indicating the strength of mutual helping in communities. A high rate of living with family and using primary care could be a strong point in rural contexts.

Integrating rural people and communities into HSB interventions could be essential for effective HSBs among older people. The ideas of rural people and communities can vary, and each community should establish an effective way of using lay and professional care through dialogue with people and healthcare professionals in communities [46,47]. They could be motivated through dialogue to act on their symptoms with effective HSBs that fit their cultural and social situations [48]. Strong mutual support in rural contexts could be integrated into effective rural HSBs among older people [49,50]. Since 2020, rural older adults have been restricted in their behaviors because of strict infection control measures and fear of COVID-19 [51,52,53,54]. For better health conditions with effective HSBs, each rural community should consider mitigating older people’s fear and ways of using lay and professional care in rural communities through dialogue with health care professionals.

This study had certain limitations. This study focused on HSBs for usual symptoms. This result may differentiate HSB preferences regarding different symptoms. As patients with specific diseases may have specific trends in HSBs, future research could focus on specific diseases to investigate preferences related to HSBs. Due to the study’s cross-sectional design, the causal relationship between intention regarding HSBs and QOL could not be determined. Interventions concerning HSB intention may be able to change QOL. However, when older people have better QOL, they feel they can confidently manage mild symptoms by themselves because they can control their usual conditions effectively. Hence, interventional studies should clarify whether interventions for HSB intentions improve QOL to address this limitation. Finally, as this study was conducted among older Japanese rural communities, the results lack generalizability to other contexts. Investigating a broader range of communities can offer a better understanding.

## 5. Conclusions

This study showed the association between HSB intention for usual symptoms and high QOL among older rural people. The association persisted after adjusting for factors previously shown to be related to QOL, including social support and capital, SES, and concrete HSBs. Enforcement of changes in older people’s HSBs without improvement in self-efficacy and the intention to perform HSBs may not change their health conditions. For better health conditions, changes in HSBs alone may be insufficient. The perception of HSBs should be investigated to improve the health conditions among older rural people. The interventions related to HSBs among older people should include a systems approach to improve self-efficacy and HSB intention by changing health systems and providing educational systems fitting their cultural context.

## Figures and Tables

**Figure 1 ijerph-19-13331-f001:**
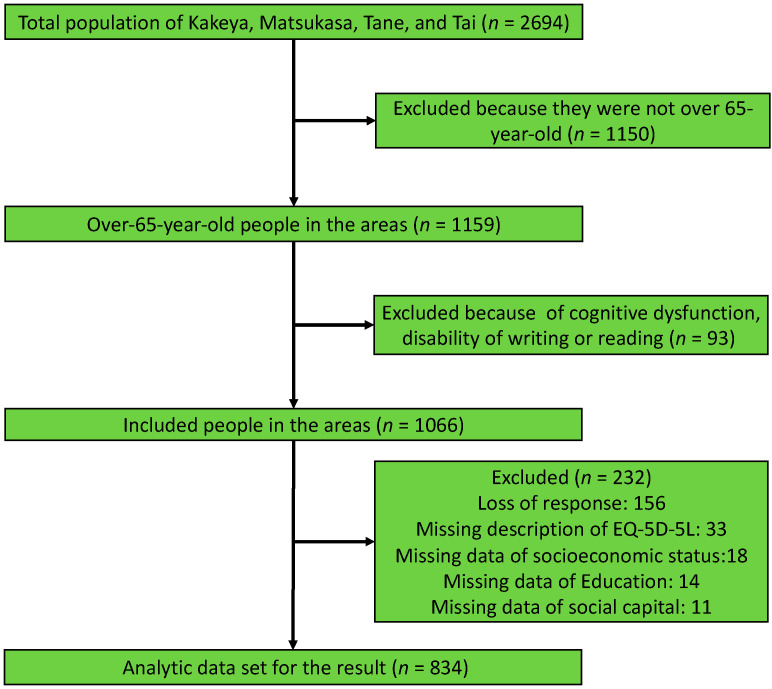
Flowchart of the participant selection process.

**Table 1 ijerph-19-13331-t001:** Demographic data of the participants in each group and the significance level of the comparisons between the two groups.

	High QOL	Low QOL	*p*-Value
Variables	*n* = 418	*n* = 416	
age (in years), mean (SD)	74.82 (7.24)	80.39 (7.78)	<0.001
sex, male (%)	191 (45.7)	175 (42.2)	0.329
BMI, mean (SD)	22.77 (3.86)	22.50 (3.71)	0.294
having chronic diseases (%)	342 (81.8)	379 (91.1)	<0.001
clinic (%)	352 (84.2)	327 (78.6)	0.041
alcohol (%)	167 (40.0)	117 (28.1)	<0.001
tobacco (%)	34 (8.1)	28 (6.7)	0.51
higher education (%)	224 (53.6)	148 (35.6)	<0.001
living with family (%)	370 (88.5)	351 (84.4)	0.086
annual health check-up (%)	326 (78.0)	270 (64.9)	<0.001
high SES (%)	267 (63.9)	173 (41.6)	<0.001
high social capital (%)	370 (88.5)	322 (77.4)	<0.001
high social support (%)	370 (88.5)	327 (78.6)	<0.001
perception of HSB			
knowledge (%)	311 (74.4)	240 (57.7)	<0.001
understanding (%)	164 (39.2)	130 (31.2)	0.017
self-efficacy (%)	97 (23.2)	48 (11.5)	<0.001
intention (%)	123 (29.4)	50 (12.0)	<0.001
concrete HSBs			
self-management (%)	272 (65.1)	219 (52.6)	<0.001
family consultation (%)	124 (29.7)	182 (43.8)	<0.001
self-medication (%)	191 (45.7)	150 (36.1)	0.005
primary care doctor (%)	282 (67.5)	315 (75.7)	0.009

Note. BMI: body mass index; Clinic: regular visit to a primary care doctor; SES: socioeconomic status; HSB: help-seeking behavior.

**Table 2 ijerph-19-13331-t002:** Associations between perceptions of help-seeking behaviors and higher quality of life.

Factors	Odds Ratio	95% CI	*p*-Value
age (in years), mean (SD)	0.9	0.88–0.92	<0.001
sex, male (%)	1.07	0.73–1.58	0.72
BMI, mean (SD)	1.03	0.98–1.08	0.27
having chronic diseases (%)	0.51	0.30–0.86	0.011
Clinic (%)	1.32	0.86–2.03	0.21
alcohol (%)	1.05	0.71–1.57	0.8
tobacco (%)	0.53	0.28–1.00	0.048
higher education (%)	0.85	0.59–1.21	0.35
living with family (%)	1.37	0.85–2.22	0.2
annual health check-up (%)	1.36	0.93–1.98	0.12
high SES (%)	2.37	1.68–3.34	<0.001
high social capital (%)	1.96	1.23–3.13	0.0046
high social support (%)	1.81	1.10–2.96	0.019
perception of HSB			
knowledge (%)	1.43	0.98–2.09	0.063
understanding (%)	0.89	0.60–1.32	0.56
self-efficacy (%)	1.66	0.99–2.78	0.053
intention (%)	2.95	1.83–4.75	<0.001
concrete HSBs			
self-management (%)	1.25	0.88–1.77	0.21
family consultation (%)	0.53	0.37–0.76	<0.001
self-medication (%)	1.33	0.94–1.88	0.11
primary care doctor (%)	0.65	0.44–0.97	0.034

Note. BMI: body mass index; Clinic: regular visit to a primary care doctor; SES: socioeconomic status: HSB, help-seeking behavior.

## Data Availability

The data presented in this study are available on request from the corresponding author.

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
