# Peer review of "Associations between Perception of Help-Seeking Behaviors and Quality of Life among Older People in Rural Communities: A Cross-Sectional Study"

_ijerph, 2022, doi:10.3390/ijerph192013331_

Round 1

Reviewer 1 Report

It's my pleasure to comment your article. The topic of your study is an interesting issue. There are some suggestions.

1) In the Section Introduction, the contribution must be offered. In addtion, the help-seeking behaviors and quality of life among elderly people in rural Japan must be introducte, and some previous literature in Japan is also offered.

2) In Section 2.1, could you tell more details why you choose  in Kakeya and Yoshida in Unnan City?

3) To  my knowledge,  you fail to the theoretical framwork. You should write the theoretical mechanism about the relationship between HSBs and QOL.  Besides, if possible, I hope you can offer more details to the social cognitive theory.

Author Response

Responses to the Reviewers’ Comments

Thank you for giving us the opportunity to submit a revised draft of our manuscript. We are grateful and appreciate the time and effort that all the reviewers have dedicated to providing valuable feedback and suggestions for its improvement. We have addressed the suggestions and highlighted the revisions in red font in this response letter and the manuscript. Here is a point-by-point response to the reviewers’ comments and concerns.

It's my pleasure to comment your article. The topic of your study is an interesting issue. There are some suggestions.

  • In the Section Introduction, the contribution must be offered. In addtion, the help-seeking behaviors and quality of life among elderly people in rural Japan must be introduce, and some previous literature in Japan is also offered.

Response:

Thank you for your valuable feedback. Per your suggestion, we have added the contribution of this study and evidence of help seeking behaviors in rural Japan as follows.

Contribution:

“Clarifying the association between the perception of HSBs and QOL among the older rural population can effectively change the focus of interventions related to HSBs from people’s behaviors to their perception of HSBs. The content of the interventions can then focus on older people’s self-efficacy and intention to performHSBs, rather than only behaviors.” (Lines 66-69)

Literature on HSB:

“According to social cognitive theory, effective health behaviors require older people to have self-efficacy and the intention to perform HSBs [26–28]. Social cognitive theory is essential for understanding HSBs through the interactions between persons, contexts, and behaviors. Self-efficacy is an individual's belief in their capacity to perform HSB successfully, leading to intention of the behaviors. Self-efficacy and intention are important prerequisites for HSBs [26–28]. Per the theory, knowledge and understanding of healthy behaviors alone may not drive effective behaviors, and high self-efficacy and intention are possible drivers of such behavior. The theory can explain the process of change in HSBs as one of the health-related behaviors [26–28].However, with respect to the social cognitive model, existing evidence does not clarify the relationship between people’s perceptions of HSBs and QOL in relation to self-efficacy and intentions. Therefore, this study aimed to answer the research question: “Is the perception of HSBs, among rural older people, associated with high QOL?” (Lines 57-66)

  • In Section 2.1, could you tell more details why you choose in Kakeya and Yoshida in Unnan City?

Response:

Thank you for the suggestion. We apologize for not providing the details in the previous version of the manuscript. We have added a detailed explanation behind choosing Kakeya and Yoshida in the method section, as follows.

“Among the six town of Unnan city, Kakeya and Yoshida lie in the extreme south-western part. They have two clinics each, and are far from a general hospital, which is more than a 30-minute drive. They are thus different from other towns which have multiple clinics and hospitals. Totally, Kakeya and Yoshida towns have six regions, among which participants from four regions (Kakeya, Tai, Matsukasa, and Tane) agreed to participate in the research.” (Lines 76-80)

  • To my knowledge, you fail to the theoretical framework. You should write the theoretical mechanism about the relationship between HSBs and QOL.  Besides, if possible, I hope you can offer more details to the social cognitive theory.

Response:

Thank you for pointing out this critical error. We have revised the introduction section to show the theoretical mechanism behind the relationship between HSBs and QOL. We have also added a comprehensive explanation of the social cognitive theory, as follows.

“Social factors such as age, gender, living location, and relationships with others in the community affects the tendency to use lay or professional care [6–8]. HSBs can affect people’s health conditions, such as self-rated health and quality of life (QOL) [6,9,10]. Hence, it is important to consider HSBs to improve healthcare“  (Page 1)

Social Cognitive Theory:

“According to social cognitive theory, effective health behaviors require older people to have self-efficacy and the intention to perform HSBs [26–28]. Social cognitive theory is essential for understanding HSBs through the interactions between persons, contexts, and behaviors. Self-efficacy is an individual's belief in their capacity to perform HSB successfully, leading to intention of the behaviors. Self-efficacy and intention are important prerequisites for HSBs [26–28]. Per the theory, knowledge and understanding of healthy behaviors alone may not drive effective behaviors, and high self-efficacy and intention are possible drivers of such behavior. The theory can explain the process of change in HSBs as one of the health-related behaviors [26–28].However, with respect to the social cognitive model, existing evidence does not clarify the relationship between people’s perceptions of HSBs and QOL in relation to self-efficacy and intentions. Therefore, this study aimed to answer the research question: “Is the perception of HSBs, among rural older people, associated with high QOL?” (Lines 57-66)

Reviewer 2 Report

1.      As reported by the authors explicitly, elderly participants whose health-seeking behaviors (HSBs thereafter) are subjected to analysis are investigated by means of established questionnaires. It is stated that the participants come from the four regions that constitute Unnan city, which in total has six regions. What is the rationale behind leaving those two regions out of the context of the study as a potential participant pool? What are the characteristics of left-out regions in terms of demography or any other factor that justifies their exclusion? [81-87]

2.     The authors utilized the health status index score as a measure for the dependent variable quality of life (QOL thereafter), for which participants’ responses to the five dimensions of the EuroQOL questionnaire are turned into single-digit health status index scores. The reference to the formula is made, but that would be quite informative to readers of the paper had the paper included the specific formula as a footnote (or endnote). [129-141]

3.     The median score is utilized as a split between elderly people with high QOL and low QOL, and statistically significant differences were investigated amongst the participants in terms of perceptions of HSBs and concrete HSBs and select confounding variables. However, that would be quite informative had the readers included a summary of the QOL variable to help readers understand its distributional properties. Perhaps, boxplots are a way to go [201-203].

4.     From Table 2, it is evident that family consultation and primary care doctor as concrete HSBs have a significant negative association with QOL. And amongst the perception of HSBs, only intention has a statistically significant impact on QOL. However, the authors concluded that “after controlling for perception regarding HSBs, concrete behaviors were not associated with QOL”. It seems that this is not the case given some of the negative associations found amongst two concrete HSBs. This inconsistency needs to be dealt with for clarity. [218-219] & [232-235]

5.     What is the authors’ opinion as to the insignificant impact that the perceptions of HSBs dimensions turned out to have on QOL other than intention? [218-219].

6.     The conclusion part needs to be developed. The result of the study should be interpreted according to other studies in the literature. Policy recommendations should be clearer.

Author Response

Responses to the Reviewers’ Comments

Thank you for giving us the opportunity to submit a revised draft of our manuscript. We are grateful and appreciate the time and effort that all the reviewers have dedicated to providing valuable feedback and suggestions for its improvement. We have addressed the suggestions and highlighted the revisions in red font in this response letter and the manuscript. Here is a point-by-point response to the reviewers’ comments and concerns.

  1. As reported by the authors explicitly, elderly participants whose health-seeking behaviors (HSBs thereafter) are subjected to analysis are investigated by means of established questionnaires. It is stated that the participants come from the four regions that constitute Unnan city, which in total has six regions. What is the rationale behind leaving those two regions out of the context of the study as a potential participant pool? What are the characteristics of left-out regions in terms of demography or any other factor that justifies their exclusion? [81-87]

Response:

Thank you for your keen observations of our study. We apologize for not being clear and for not providing the details. Participants from only four of the regions had agreed to participate in the research. Hence, the other two regions could not be a part of the study. We have added this explanation in the manuscript. We have also included details regarding the characteristics of these regions, as shown below.  

“Among the six town of Unnan city, Kakeya and Yoshida lie in the extreme south-western part. They have two clinics each, and are far from a general hospital, which is more than a 30-minute drive. They are thus different from other towns which have multiple clinics and hospitals. Totally, Kakeya and Yoshida towns have six regions, among which participants from four regions (Kakeya, Tai, Matsukasa, and Tane) agreed to participate in the research.” (Lines 76-80)

  1. The authors utilized the health status index score as a measure for the dependent variable quality of life (QOL thereafter), for which participants’ responses to the five dimensions of the EuroQOL questionnaire are turned into single-digit health status index scores. The reference to the formula is made, but that would be quite informative to readers of the paper had the paper included the specific formula as a footnote (or endnote). [129-141] 

Response:

Thank you for your valuable suggestion. We reviewed the other articles regarding description of EQ-5D-5L. There are few papers that describe the formula specifically, because of the complexity of the formula. Therefore, we were not able to include it.

  1. The median score is utilized as a split between elderly people with high QOL and low QOL, and statistically significant differences were investigated amongst the participants in terms of perceptions of HSBs and concrete HSBs and select confounding variables. However, that would be quite informative had the readers included a summary of the QOL variable to help readers understand its distributional properties. Perhaps, boxplots are a way to go [201-203]

Response:

Thank you for your suggestions. Per your suggestion, we have added the interquartile range of EQ-5D-5L score to help readers understand its distributional properties as follows.

“Parametric and categorical data were analyzed using Student’s t-test and chi-squared test, respectively. A significance level of p < 0.05 was used for all comparisons. Participants were divided into two groups depending on whether their health status index scores were above or below the median. The median was 0.705, with an interquartile range of 0.592 to 0.814." (Lines 132-135)

  1. From Table 2, it is evident that family consultation and primary care doctor as concrete HSBs have a significant negative association with QOL. And amongst the perception of HSBs, only intention has a statistically significant impact on QOL. However, the authors concluded that “after controlling for perception regarding HSBs, concrete behaviors were not associated with QOL”. It seems that this is not the case given some of the negative associations found amongst two concrete HSBs. This inconsistency needs to be dealt with for clarity. [218-219] & [232-235]

Response:

We apologize for the discrepancies and thank you for bringing them to our notice. We have revised the discussion by focusing on the logic of changing HSBs with aim of improving health conditions based on social cognitive theory. In addition, accordingly we have revised the conclusion as follows.

“Previous studies suggest that changes in HSBs among older people could effectively improve their health conditions [23]. However, this study revealed that after controlling for perceptions regarding HSBs, concrete behaviors were not associated with QOL. This result indicates that the concrete HSBs based on people’s perceptions of how they can behave when experiencing certain symptoms is essential. This can be explained by the social cognitive theory. According tothe theory, imposed behaviors and help may not change people’s health conditions. Rather self-efficacy and intention of health behaviors should drive people’s behaviors and are essential [26-28]. Changes in HSBs alone may not change older people’s health conditions. Therefore, enforcing changes in their HSBs without improving their self-efficacy and HSB intentions, may not change their health conditions. According to the social cognitive model concerning modification of human behaviors, older adults should be motivated to act based on their perceptions of knowledge, understanding, self-efficacy, and intentions [26].

There is a possibility of the positive relationship between intention to perform HSB and QOL. As this study shows, the intentions to perform HSBs such as approaches to usual symptoms could be associated with high QOL after adjusting for other social and health conditions. Therefore, the relationships among them can be considered in various ways. People with high QOL could have high motivation and intention regarding HSBs because they can move and act without subjective difficulty in their lives and have high social capital and support, and SES [36–38]. However, this study showed that the association between high intention to perform HSBs and QOL persisted after adjustment for these social and health conditions. Based on behavioral theories, the perception of health conditions, self-efficacy, and confidence in HSBs could lead to high QOL [39,40]. Although this study only showed an association among intention to perform HSBs and QOL, it suggests positive relationship between HSB intention and high QOL based on behavioral theories. Lines (193-213)

In particular, their intention regarding HSBs to treat usual symptoms could be associated with QOL after adjusting for other social and health conditions, showing a possible relationship. This study shows an association between them; therefore, the relationships among them can be considered in various ways. People with high QOL could have high motivation and intention regarding HSBs because they can move and act without subjective difficulty in their lives and have high social capital and support, and socioeconomic status [36–38]. However, as this study shows, the association between high intention regarding HSBs and QOL persisted after adjustment for these social and health conditions. Logically, the perception of health conditions, self-efficacy, and confidence in HSBs could lead to high QOL [39,40]. Although this study only shows an association, it suggests a relationship between HSB intention and high QOL.” (Line 187-204)

“This study showed the association between HSB intention for usual symptoms and high QOL among older rural people. The association persisted after adjusting for factors previously shown to be related to QOL, including social support and capital, SES, and concrete HSBs. Enforcement of changes in older people’s HSBs without improvement in self-efficacy and the intention to perform HSBs, may not change their health conditions. For better health conditions, changes in HSBs alone may be insufficient. The perception of HSBs should be investigated to improve the health conditions among older rural people. The interventions related to HSBs among older people should include a systems approach to improve self-efficacy and HSB intention by changing health systems and providing educational systems fitting their cultural context.” (Lines 243-250)

  1. What is the authors’ opinion as to the insignificant impact that the perceptions of HSBs dimensions turned out to have on QOL other than intention? [218-219]. 

Response:

Thank you for your query. We consider that people with high QOL have not only self-efficacy and intention to HSBs to their usual symptoms, and people with high self-efficacy and intention to HSBs may have high QOL because of less anxieties when having symptoms. We have added the ideas in the manuscript as follows.

“There is a possibility of the positive relationship between intention to perform HSB and QOL. As this study shows, the intentions to perform HSBs such as approaches to usual symptoms could be associated with high QOL after adjusting for other social and health conditions. Therefore, the relationships among them can be considered in various ways. People with high QOL could have high motivation and intention regarding HSBs because they can move and act without subjective difficulty in their lives and have high social capital and support, and SES [36–38]. However, this study showed that the association between high intention to perform HSBs and QOL persisted after adjustment for these social and health conditions. Based on behavioral theories, the perception of health conditions, self-efficacy, and confidence in HSBs could lead to high QOL [39,40]. Although this study only showed an association among intention to perform HSBs and QOL, it suggests positive relationship between HSB intention and high QOL based on behavioral theories.” (Line 207 to 216)

  1. The conclusion part needs to be developed. The result of the study should be interpreted according to other studies in the literature. Policy recommendations should be clearer.

Response:

We apologize if our conclusion was unclear. We have revised the conclusion including the interpretation of the results and policy recommendations as follows.

“This study showed the association between HSB intention for usual symptoms and high QOL among older rural people. The association persisted after adjusting for factors previously shown to be related to QOL, including social support and capital, SES, and concrete HSBs. Enforcement of changes in older people’s HSBs without improvement in self-efficacy and the intention to perform HSBs, may not change their health conditions. For better health conditions, changes in HSBs alone may be insufficient. The perception of HSBs should be investigated to improve the health conditions among older rural people. The interventions related to HSBs among older people should include a systems approach to improve self-efficacy and HSB intention by changing health systems and providing educational systems fitting their cultural context.” (Lines 240 to 247)

Round 2

Reviewer 1 Report

Thank you for your response to my review comments. In my opinion, your manuscript should be accepted.